# Extended-spectrum beta-lactamase (ESBL)-producing and non-ESBL-producing *Escherichia coli* isolates causing bacteremia in the Netherlands (2014 – 2016) differ in clonal distribution, antimicrobial resistance gene and virulence gene content

**Denise van Hout**[1]*, **Tess D. Verschuuren**[1], **Patricia C. J. Bruijning-Verhagen**[1,2], **Thijs Bosch**[2], **Anita C. Schürch**[3], **Rob J. L. Willems**[3], **Marc J. M. Bonten**[1,2,3], **Jan A. J. W. Kluytmans**[1,4]

1 Julius Center for Health Sciences and Primary Care, University Medical Center Utrecht, University Utrecht, Utrecht, The Netherlands, 2 Center for Infectious Disease Control, National Institute for Public Health and the Environment, Bilthoven, The Netherlands, 3 Department of Medical Microbiology, University Medical Center Utrecht, University Utrecht, Utrecht, The Netherlands, 4 Microvida Laboratory for Medical Microbiology and Department of Infection Control, Amphia Hospital, Breda, The Netherlands

* D.vanHout-3@umcutrecht.nl

## Abstract

### Background

Knowledge on the molecular epidemiology of *Escherichia coli* causing *E. coli* bacteremia (ECB) in the Netherlands is mostly based on extended-spectrum beta-lactamase-producing *E. coli* (ESBL-Ec). We determined differences in clonality and resistance and virulence gene (VG) content between non-ESBL-producing *E. coli* (non-ESBL-Ec) and ESBL-Ec isolates from ECB episodes with different epidemiological characteristics.

### Methods

A random selection of non-ESBL-Ec isolates as well as all available ESBL-Ec blood isolates was obtained from two Dutch hospitals between 2014 and 2016. Whole genome sequencing was performed to infer sequence types (STs), serotypes, acquired antibiotic resistance genes and VG scores, based on presence of 49 predefined putative pathogenic VG.

### Results

ST73 was most prevalent among the 212 non-ESBL-Ec (N = 26, 12.3%) and ST131 among the 69 ESBL-Ec (N = 30, 43.5%). Prevalence of ST131 among non-ESBL-Ec was 10.4% (N = 22, *P* value < .001 compared to ESBL-Ec). O25:H4 was the most common serotype in both non-ESBL-Ec and ESBL-Ec. Median acquired resistance gene counts were 1 (IQR 1–6) and 7 (IQR 4–9) for non-ESBL-Ec and ESBL-Ec, respectively (*P* value < .001). Among non-ESBL-Ec, acquired resistance gene count was highest among blood isolates from a

**Data Availability Statement:** All relevant data are within the manuscript and its Supporting Information files.

**Funding:** The authors received no specific funding for this work.

**Competing interests:** The authors have declared that no competing interests exist.

primary gastro-intestinal focus (median 4, IQR 1–8). Median VG scores were 13 (IQR 9–20) and 12 (IQR 8–14) for non-ESBL-Ec and ESBL-Ec, respectively (*P* value = .002). VG scores among non-ESBL-Ec from a primary urinary focus (median 15, IQR 11–21) were higher compared to non-ESBL-Ec from a primary gastro-intestinal (median 10, IQR 5–13) or hepatic-biliary focus (median 11, IQR 5–18) (*P* values = .007 and .04, respectively). VG content varied between different *E. coli* STs.

## Conclusions

Non-ESBL-Ec and ESBL-Ec blood isolates from two Dutch hospitals differed in clonal distribution, resistance gene and VG content. Also, resistance gene and VG content differed between non-ESBL-Ec from different primary foci of ECB.

## Introduction

*Escherichia coli* is the leading causative pathogen in Gram-negative bacteremia and is associated with 30-day mortality up to 18% [1–4]. Antibiotic treatment options of *E. coli* bacteremia (ECB) are getting compromised by the pandemic presence of extended-spectrum beta-lactamases (ESBLs); conferring resistance to antibiotics commonly used for ECB treatment such as third-generation cephalosporins. Worryingly, the incidence of ECB is increasing and in some European countries, the incidence of ECB with antibiotic-resistant strains seems to increase faster than ECB caused by susceptible strains [3–6]. Even though the individual patient and financial burden is increased for resistant ECB episodes, ECB due to susceptible strains is far more common and therefore determines the major part of the ECB disease burden. The majority of ECBs is of community onset and is preceded by an infection in the urinary tract, but other sources, such as the hepatic-biliary tract, also comprise important primary foci [3,7]. These clinical characteristics of ECB episodes are important because they indicate different target populations for prevention. Thorough insight in the molecular epidemiology of both ESBL-negative and ESBL-positive ECB episodes with different clinical characteristics is key in identifying targets for the development of future preventive strategies, such as *E. coli* vaccines that are currently being developed [8]. Up to now, the molecular epidemiology of ECB in the Netherlands was mainly described in single-center studies [9] or among antimicrobial resistant isolates only [10].

In this study, we aimed to analyze the current population structure of ECB in the Netherlands, with special attention to differences in antimicrobial resistance and virulence gene (VG) content and clonal and serotype distribution between isolates with different clinical epidemiological characteristics and between non-ESBL-producing *E. coli* (non-ESBL-Ec) and ESBL-producing *E. coli* (ESBL-Ec) blood isolates.

## Methods

### Study design

Details of the study design is fully described elsewhere [11]. In short, unique patients with ECB were retrospectively identified in the University Medical Center Utrecht, a 1,042-bed tertiary care center and the Amphia Hospital in Breda, an 837-bed teaching hospital. In each hospital, a random sample of 40 isolates per year for the years 2014, 2015 and 2016 was selected, comprising ~24% of all first bacteremic *E. coli* isolates in a year. In addition, all ESBL-Ec blood

isolates from 2014 to 2016 were selected. Whole genome sequencing (WGS) was performed by The Netherlands National Institute for Public Health and the Environment (RIVM) using the Illumina HiSeq 2500 (BaseClear, Leiden, the Netherlands). All generated raw reads were submitted to the European Nucleotide Archive (ENA) of the European Bioinformatics Institute (EBI) under the study accession number PRJEB35000. De novo assembly was performed using SPAdes genome assembler v.3.6.2 and quality of assembles was assessed using QUAST [12]. ESBL-production was defined as confirmed phenotypic ESBL-positivity, unless described otherwise [11]. Baseline characteristics were compared between non-ESBL-Ec and ESBL-Ec ECB episodes by the Fisher's Exact or Pearson $\chi^2$ test for categorical variables and by Mann-Whitney U test for continuous variables when applicable. A two-tailed *P* value < .05 was considered statistically significant.

This study does not fall under the scope of the Medical Research Involving Human Subjects Act. The Medical Research Ethics Committee of the UMCU has therefore waived the need for official approval by the Ethics Committee (IRB number 18/056). Individual informed consent was not obtained and all study data were analyzed and stored in a pseudonymized form. All statistical analyses were performed with Statistical Package for Social Sciences V.25.0 (SPSS, Chicago, Illinois, USA) and R Version 3.4.1.

## Multi-locus sequence types (MLST)

Multi-locus sequence types (STs) were based on the allelic profile of seven housekeeping genes and were determined using mlst2.0 (https://github.com/tseemann/mlst), by scanning contig files against the *E. coli* PubMLST typing scheme (updated May 12th, 2018). Clonal (i.e. ST) distribution was presented stratified for non-ESBL-Ec and ESBL-Ec isolates and by epidemiological subgroups. Genotype (ST) diversity was analysed by Simpson's diversity index [13].

## Serotyping

Serotypes were assigned by using the web-tool SerotypeFinder 2.0 from the Center for Genomic Epidemiology at the Danish Technical University, Lyngby, Denmark (https://cge.cbs.dtu.dk/services/SerotypeFinder) [14]. Simpson's index for serotype diversity was calculated for non-ESBL-Ec and ESBL-Ec isolates. Serotype distribution among non-ESBL-Ec and ESBL-Ec was compared to two current *E. coli* vaccine candidates [8,15], excluding those isolates in which no definitive serotype could be defined.

## Antimicrobial resistance genes and virulence genes

Abricate (https://github.com/tseemann/abricate) v0.8.13 was used for (i) mass screening of contigs for (acquired) antimicrobial resistance genes using ResFinder 3.1.0 (download 24 January 2019), and (ii) to determine presence of VG by BLAST against the VFDB database (http://www.mgc.ac.cn/VFs) (download 8 February 2019)[16,17]. We searched for 49 putative VG that were previously described as extra-intestinal pathogenic *E. coli* (ExPEC)-associated VG [18–22]. If any of the predefined VG were not included in VFDB, BLAST against the ecoli_VF_collection database was performed (date 8 February 2019)[23]. Coverage length and sequence identity thresholds were 80% and 95%. Resistance gene count was defined as the total number of unique identified acquired resistance genes per isolate. Resistance gene counts were compared between non-ESBL-Ec and ESBL-Ec with the non-parametric Wilcoxon rank sum test (for this comparison only, resistance gene count of ESBL-Ec was corrected for presence of the ESBL gene). The VG score was defined as the total number of pre specified VG within an isolate, adjusted for multiple detection of the *afa/dra* (Afa/Dr adhesins), *pap* (P fimbrial adhesins), *sfa/foc* (S and F1C fimbrial adhesins) and *kpsM* (group 2 and III capsule)

operons, as described previously [20]. If a VG was detected multiple times within a single isolate (i.e. different quality measures), it was only counted once. The *kpsM*, *afa/dra* and *sfa/foc* operons were considered present if any of the corresponding genes or allelic variants were identified.

Resistance gene counts and VG scores were further analysed for non-ESBL-Ec and ESBL-Ec separately and were compared between isolates with different epidemiological characteristics and different STs using Kruskal-Wallis one-way ANOVA. In case of an overall ANOVA *P* value < .05, post-hoc pairwise comparisons were made with the non-parametric Wilcoxon-rank sum test and the Holm-Bonferroni *P* value correction was applied to account for multiple testing.

## Results

### Patient characteristics

The isolate collection consisted of 212 phenotypic non-ESBL-Ec and 69 ESBL-Ec blood isolates (Fig 1). Distribution of age, sex, onset of infection and primary foci were comparable between non-ESBL-Ec and ESBL-Ec bacteremia episodes (Table 1). As compared to non-ESBL-Ec, ECB episodes with ESBL-Ec were less often of community onset (63.8% versus 81.1%, *P* value = .003). Crude 30-day and 1-year mortality were higher in ECB episodes caused by ESBL-Ec (27.5% and 50.7%, respectively) compared to ECB episodes caused by non-ESBL-Ec (11.3% and 29.2%, respectively) (both *P* values = .001).

### Clonal distribution

Among non-ESBL-Ec, ST73 was the most frequently observed ST (N = 26, 12.3%), followed by ST131 (N = 22, 10.4%). Isolates of ST73, 95, 127, 141, 80 and 1193 were solely identified among non-ESBL-Ec (Fig 2). ST131 was dominant among ESBL-Ec (N = 30, 43.5%) and prevalence was higher than among non-ESBL-Ec (*P* value < .001). Simpson's index for clonal diversity was 95.6% (95% CI 94.4% – 96.8%) and 80.6% (95% CI 70.9% – 90.4%) for non-ESBL-Ec and ESBL-Ec, respectively. The occurrence of different STs did not differ between nosocomial and community onset ECB (S1 Appendix). ST131 was the dominant ST among ESBL-positive ECB episodes with a primary urinary (63%) and gastro-intestinal focus (57%), which was higher as compared to other primary foci of ESBL-positive ECB (i.e. 21% among primary hepatic-biliary focus, S1 Appendix).

### Serotypes

The most common serotype O25:H4 was identified in 19 (9.0%) non-ESBL-Ec and 24 (34.8%) ESBL-Ec isolates, which largely reflected the prevalence of ST131 in each group (Table 2). Multiple serotypes only occurred among non-ESBL-Ec, such as O6:H1 and O6:H31. ST73 was most often of serotype O6:H1 (16/26, 61.5%). Simpson's index for serotype diversity was 96.7% (95% CI 95.8% – 97.6%) and 83.8% (95% CI 76.9% – 90.6%) for non-ESBL-Ec and ESBL-Ec, respectively. Non-ESBL-Ec and ESBL-Ec isolates from ECB episodes with a primary focus in the urinary tract were most often of O-serotype O6 (15/103, 14.6%) and O25 (17/30, 56.7%), respectively (S2 Appendix). For ECB episodes with a primary focus in the hepatic-biliary tract, O25 was the most prevalent O-serotype among non-ESBL-Ec (7/46, 15.2%) and O8 (4/14, 28.6%) among ESBL-Ec isolates (S2 Appendix).

53 (25.0%) non-ESBL-Ec and 25 (36.2%) ESBL-Ec isolates belonged to either O1, O2, O6 or O25, the serotypes of the 4-valent *E. coli* vaccine that has reached phase 2 development stage [8,24], whereas the majority of non-ESBL-Ec (N = 113; 53.3%) and ESBL-Ec isolates (N = 35;

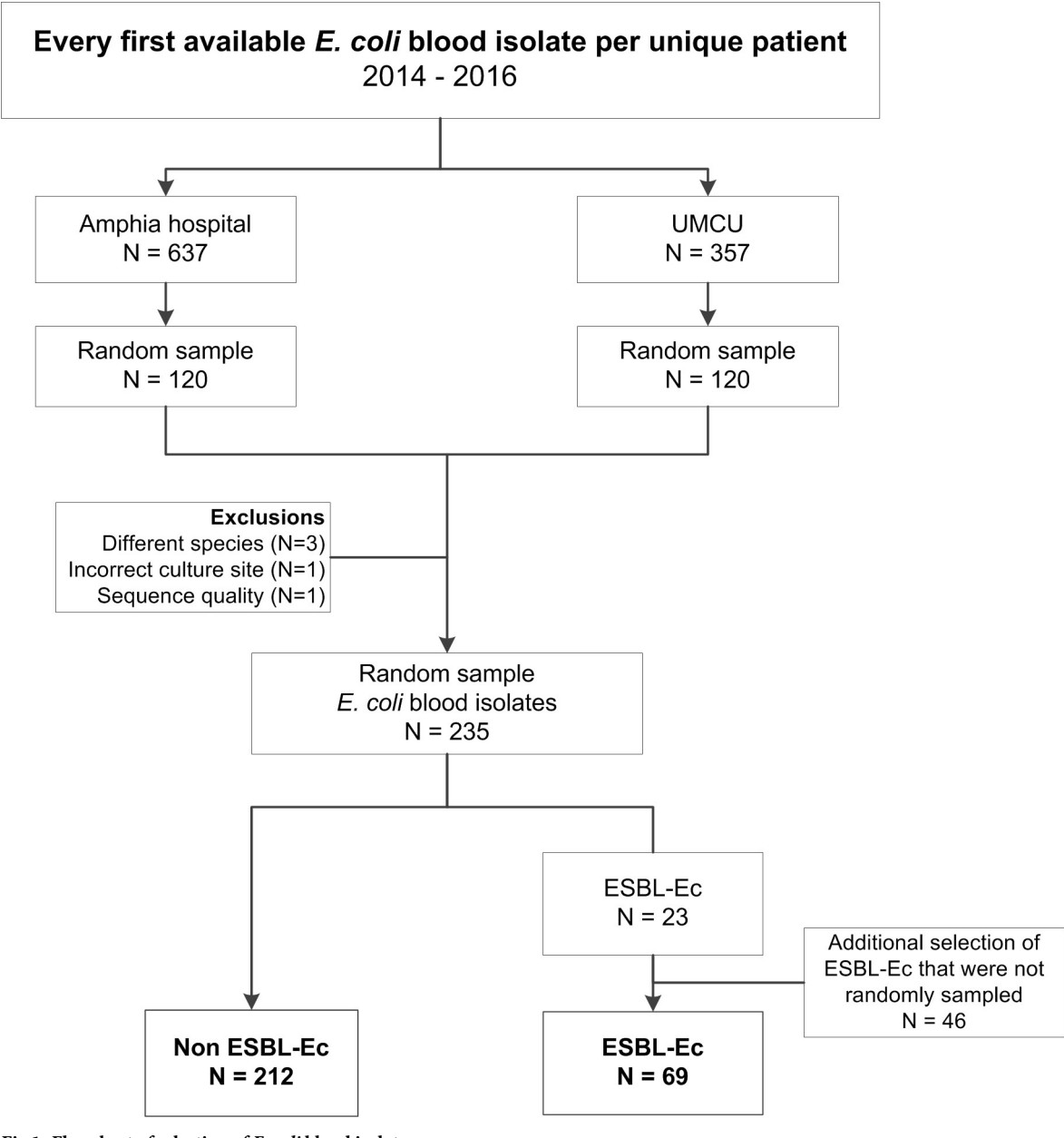

**Fig 1. Flowchart of selection of *E. coli* blood isolates.**

50.7%) belonged to one of the O-serotypes of the new 10-valent conjugant *E. coli* vaccine (ExPEC-10V) that is currently in development [15].

## Antimicrobial resistance genes

In total, 69 unique acquired resistance genes were identified (S3 Appendix). ESBL-genes were detected in 65 (94.2%) of 69 *E. coli* blood isolates with phenotypic ESBL production. $bla_{CTX-M-15}$ was most prevalent (N = 28, 43.1%), followed by $bla_{CTX-M-9}$ (N = 14, 21.5%) and $bla_{CTX-M-27}$ (N = 9, 13.8%). The median acquired resistance gene count for non-ESBL-Ec versus ESBL-Ec was 1 (IQR 1–6) versus 7 (IQR 4–9) ($P$ value < .001).

**Table 1. Baseline epidemiological characteristics of *E. coli* bacteremia episodes.**

| | Non-ESBL-Ec[a] N = 212 | | ESBL-Ec[a] N = 69 | | P value[b] |
|---|---|---|---|---|---|
| Median age, years (IQR) | 69 | (59–77) | 69 | (56–76) | .80 |
| Female sex (%) | 102 | (48.1) | 32 | (46.4) | .80 |
| Community onset (%) | 172 | (81.1) | 44 | (63.8) | *.003* |
| Primary focus of ECB (%) | 103 | (48.6) | 30 | (43.5) (20.3) (10.1) | .79 |
| Urinary tract | 46 | (21.7) | 14 | (7.2) | |
| Hepatic-biliary | 23 | (10.8) | 7 | (18.8) | |
| Gastro-intestinal | 10 | (4.7) | 5 | | |
| Other | 30 | (14.2) | 13 | | |
| Unknown | | | | | |
| Urinary catheter (%) | 69 | (32.5) | 28 | (40.6) | .22 |
| Ward (%) | 182 | (85.8) | 58 | (84.1) (15.9) | .71 |
| Non-ICU | 30 | (14.2) | 11 | | |
| ICU | | | | | |
| Mortality (%) | 24 | (11.3) | 19 | (27.5) (50.7) | *.001* |
| 30-day | 62 | (29.2) | 35 | | *.001* |
| 1-year | | | | | |

ECB, *E. coli* bacteremia; ESBL, extended-spectrum beta-lactamase; ESBL-Ec, ESBL-producing *E. coli*; ICU, intensive care unit; IQR, interquartile range; non-ESBL-Ec, non-ESBL-producing *E. coli*.

[a]ESBL-positivity based on phenotype.

[b]P value of comparison between non-ESBL-Ec versus ESBL-Ec, calculated with Pearson's χ2, Fisher's exact, or Mann-Whitney U test when applicable. P values in italic represent P values < .05.

Among non-ESBL-Ec, acquired resistance gene count was highest among blood isolates from a primary gastro-intestinal focus (median 4, IQR 1–8). There were significant differences in resistance gene count for different primary foci of non-ESBL ECB, but absolute differences were small (S3 Appendix). Among ESBL-Ec isolates, there were no statistical significant

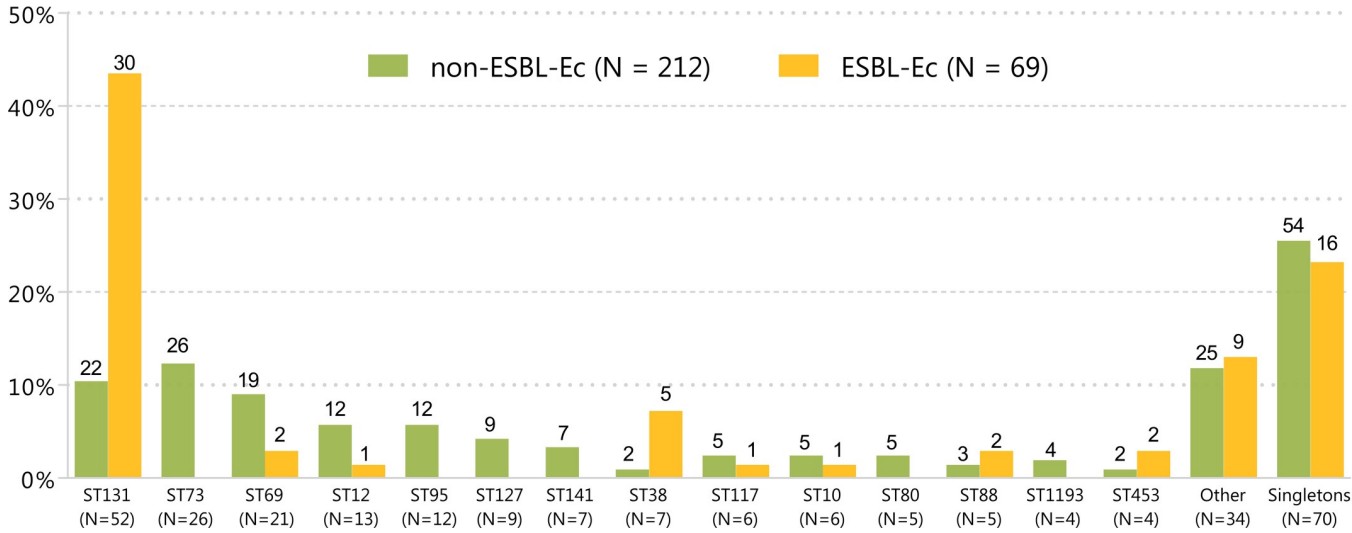

**Fig 2. ST distribution among non-ESBL-Ec versus ESBL-Ec[a] in order of frequency[b].** ESBL, extended-spectrum beta-lactamase; ESBL-Ec, ESBL-producing *E. coli*; non-ESBL-Ec, non-ESBL-producing *E. coli*; ST, sequence type. [a]ESBL-positivity based on phenotypic ESBL production. [b]Missing STs and STs that occurred ≤3 times are grouped in "Other". STs that only occurred once are grouped in "Singletons". The height of each individual bars represents the proportion of the ST within the group of non-ESBL-Ec and ESBL-Ec, respectively. The numbers represent the absolute numbers of occurrence.

**Table 2. Serotype distribution among *E. coli* blood isolates, stratified for ESBL-positivity.**

| | Non-ESBL-Ec N = 212 (%) | ESBL-Ec[a] N = 69 (%) |
|---|---|---|
| O25:H4 | 19 (9.0) | 24 (34.8) |
| O6:H1 | 16 (7.5) | - |
| O2/O50:H6 | 10 (4.7) | - |
| O6:H31 | 9 (4.2) | - |
| O15:H18 | 7 (3.3) | 2 (2.9) |
| O17/O44/O77:H18 | 8 (3.8) | - |
| O4:H5 | 7 (3.3) | 1 (1.4) |
| O75:H5 | 8 (3.8) | - |
| O8:H9 | 5 (2.4) | 2 (2.9) |
| O16:H5 | 3 (1.4) | 3 (4.3) |
| O86:H18 | 1 (0.5) | 4 (5.8) |
| O4:H1 | 5 (2.4) | - |
| O1:H7 | 4 (1.9) | - |
| O117:H4 | 4 (1.9) | - |
| O2/O50:H1 | 4 (1.9) | - |
| O23:H16 | 2 (0.9) | 2 (2.9) |
| O25:H1 | 4 (1.9) | - |
| O18/O18ac:H7 | 3 (1.4) | - |
| O2/O50:H7 | 3 (1.4) | - |
| O45:H7 | 3 (1.4) | - |
| O75:H7 | 3 (1.4) | - |
| O8:H17 | 3 (1.4) | - |
| O9:H17 | - | 2 (2.9) |
| O9/O104:H9 | - | 2 (2.9) |
| O13/O135:H4 | 2 (0.9) | - |
| O18:H1 | 2 (0.9) | - |
| O18:H5 | 2 (0.9) | - |
| O22:H1 | 2 (0.9) | - |
| O24:H4 | 2 (0.9) | - |
| O8:H10 | 2 (0.9) | - |
| O8:H25 | 2 (0.9) | - |
| O8:H30 | 2 (0.9) | - |
| Singletons | 45 (21.2) | 13 (18.8) |
| Unknown | 20 (9.4) | 14 (20.3) |

ESBL, extended-spectrum beta-lactamase; ESBL-Ec, ESBL-producing *E. coli*, non-ESBL-Ec, non-ESBL-producing *E. coli*.

[a]ESBL-positivity based on phenotypic ESBL production.

differences in acquired resistance gene counts between epidemiological subgroups (S3 Appendix). We observed no significant differences among non-ESBL-Ec or ESBL-Ec isolates of different clonal backgrounds (Fig 3 and S3 Appendix).

## Virulence genes

Of the 49 predefined ExPEC-associated VG, 44 (89.8%) were detected in at least one *E. coli* blood isolate and VG scores ranged from zero (N = 1 non-ESBL-Ec) to 25 (N = 2 ESBL-Ec) (S4

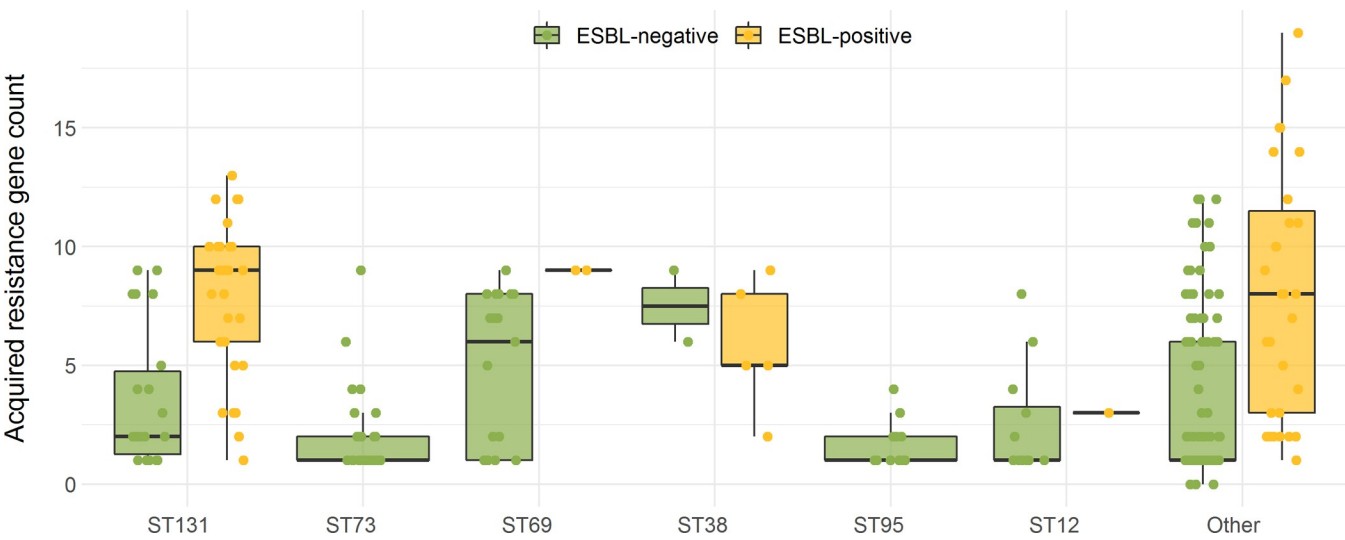

**Fig 3. Acquired resistance gene count per ST, stratified for ESBL-positivity[a].** ESBL, extended-spectrum beta-lactamase; ESBL-Ec, ESBL-producing *E. coli*; non-ESBL-Ec, non-ESBL-producing *E. coli*; ST, sequence type. [a]ESBL-positivity based on phenotypic ESBL production. Boxplots display median resistance gene count and inter quartile range (IQR); every dot represents a single isolate. Only STs that occurred >5% within non-ESBL-Ec or ESBL-Ec were grouped into main groups, the rest was categorized as "Other". Results of the pairwise comparisons between STs can be found in S3 Appendix.

Appendix). The median VG score was 13 (IQR 9–20) in non-ESBL-Ec and 12 (IQR 8–14) in ESBL-Ec blood isolates (*P* value = .002). There were no significant differences in VG scores of epidemiological subgroups, except that the average VG score of non-ESBL-Ec isolates with a primary urinary focus (median 15, IQR 11–21) were higher compared to non-ESBL-Ec isolates with a primary focus in the gastro-intestinal (median 10, IQR 5–13) or hepatic-biliary tract (median 11, IQR 5–18) (*P* values = .007 and .04, respectively) (S4 Appendix).

There was heterogeneity in VG scores between non-ESBL-Ec of different STs, this was less pronounced for ESBL-Ec isolates (Fig 4 and S4 Appendix). ESBL-negative ST38 had the lowest average VG score (median 7, IQR 6–7) and ESBL-positive ST12 had the highest VG score (median 23, IQR 23–23). Median VG score of both ESBL-negative and ESBL-positive ST131 isolates was 13 (IQR 12–15).

## Discussion

In this study, we found that ESBL-producing *E. coli* blood isolates were different from non-ESBL-producing *E. coli* causing bacteraemia in terms of clonal distribution, serotype distribution, antimicrobial resistance gene count and VG scores.

In line with previous research, the clonal distribution among ESBL-Ec blood isolates was less diverse compared to non-ESBL-Ec [25–27]. This was mainly caused by the predominance of ST131 within ESBL-Ec, as has been described before [28,29]. In contrast, ST73, a ST that so far is known for its susceptibility to antibiotics [28], was only identified among non-ESBL-Ec blood isolates. Previous studies have shown very different phylogeny of ST73 and ST131, with the first being characterised by a higher level of diversification in to divergent clades [28,30]. The association between ESBL phenotype and STs in *E. coli*, which is repeatedly found, implies that the genetic make-up of strains contributes to the ability to acquire and subsequently maintain plasmids carrying ESBL genes. Indeed, a recent large-scale study that compared the pan-genomes of invasive *E. coli* isolates, including ST131 and ST73, suggested that due to ongoing adaptation to long term human intestinal colonisation and consequent evolutionary gene

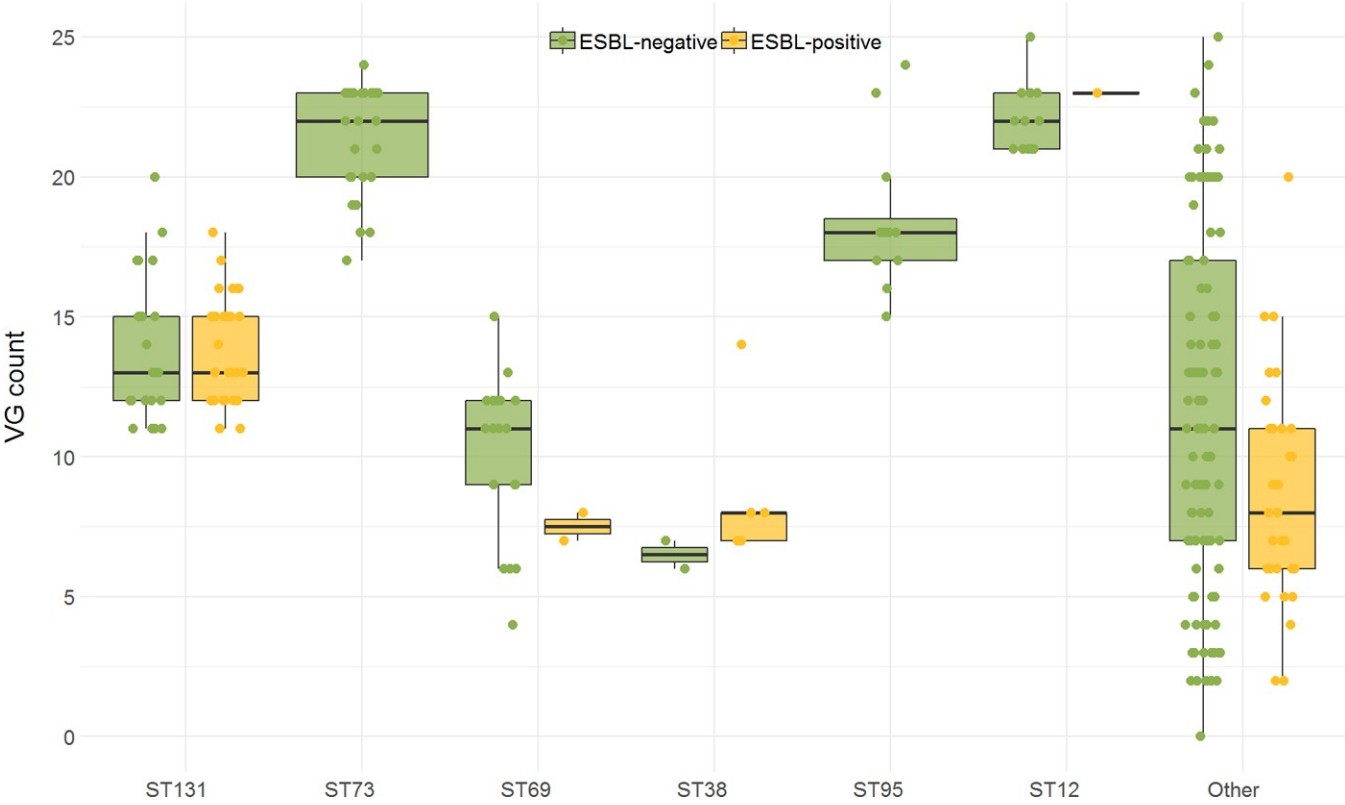

**Fig 4. ExPEC-associated VG score in different STs, stratified for ESBL-positivity[a].** ESBL, extended-spectrum beta-lactamase; ESBL-Ec, ESBL-producing *E. coli*; ExPEC, extra-intestinal pathogenic *E. coli;* non-ESBL-Ec, non-ESBL-producing *E. coli*; ST, sequence type; VG, virulence gene. [a]ESBL-positivity based on phenotypic ESBL production. Boxplots display median VG score and inter quartile range (IQR); every dot represents a single isolate. Only STs that occurred >5% within non-ESBL-Ec or ESBL-Ec were grouped into main groups, the rest was categorized as "Other". Results of pairwise comparisons between STs can be found in S4 Appendix.

selection, ST131 might have become able to reduce the fitness costs of long term plasmid maintenance [31,32]. It has been hypothesized that this is also true for other *E. coli* lineages that are associated with multidrug resistance (MDR). Reducing the fitness costs of replicating plasmids encoding MDR will result in having competitive advantage over other intestinal strains [33].

We hypothesized that the clonal distribution and resistance gene and VG content would differ between ECB episodes of community and hospital onset and between different primary foci, as a result of adaptive evolution of intestinal *E. coli*. We observed some statistical significant differences in resistance gene count and VG scores among non-ESBL-Ec from different primary foci of ECB, such as higher VG scores of blood isolates from a primary urinary focus as compared to isolates from a primary focus in the gastro-intestinal or hepatic-biliary tract. However, absolute differences in gene counts were small and the clinical significance remains unclear. In the current study, we found that differences in molecular content mostly depended on phenotypic ESBL-production and STs. This confirms the findings from a recent study that was performed in Scotland [34]. In that study, there were combinations of VGs as well as a particular accessory gene composition that differentiated between STs rather than between epidemiological factors. The association between ST69 and community onset ECB, as found in the Scottish study, was not identified in the current study. Other differences were the large proportion of *E. coli* isolates from ECB episodes that were deemed hospital-acquired (62%) as

compared to our study (18.4% for ESBL-negative and 36.2% for ESBL-positive ECB) and in that study, analyses were not stratified for ESBL-positivity.

Interestingly, in our study, isolates that belonged to ST73 had low resistance gene content but relatively high VG scores as compared to other STs. Furthermore, the average VG score among non-ESBL-Ec was slightly higher than among ESBL-Ec blood isolates, which supports findings of other studies that described an inverse association between antimicrobial resistance and VG content in ExPEC *E. coli* [35–40]. This historical negative association has been challenged, considering the current predominance of ST131, with its relatively broad VG profile despite being associated with MDR [41–43]. Also in our study, ESBL-positive and ESBL-negative ST131 isolates had equal average VG scores.

We identified serotype O25:H4 as the most prevalent serotype causing ESBL-negative as well as ESBL-positive ECB in the Netherlands, followed by O6:H1. The serotype distribution among non-ESBL-Ec was more heterogeneous compared to ESBL-Ec, similar to the differences in clonal diversity [44]. A large recent European surveillance study that included 1,110 *E. coli* blood isolates from adults between 2011 and 2017 showed that there is heterogeneity in serotype distribution among different countries, which highlights the need for country specific data, such as provided in the current study [15]. We showed that the coverage of the new potential 10-valent vaccine was higher compared to the 4-valent vaccine and was actually doubled for non-ESBL-Ec bacteraemia. Findings of the current study may help further evaluation and implementation of *E. coli* vaccines.

Strengths of the current study are the multicenter design and combination of epidemiological characteristics with highly discriminatory genetic data. There are also important limitations. Firstly, *E. coli* is a heterogeneous species, of which the seven MLST genes only constitute a small proportion of the entire gene content. Because we also only investigated a small fraction of the genes that are commonly part of the accessory genome, such as VGs and acquired resistance genes, we may have missed genomic differences that could have importantly contributed to ecological specialization in the different clinically relevant primary foci. Secondly, we selected *E. coli* isolates from a tertiary care center and teaching hospital from two different regions, which we considered to be representative of the Netherlands. The description of strains that were identified here might not be entirely generalizable to other countries since there could be differences between circulating *E. coli* strains, dependent on local population characteristics and resistance levels. Thirdly, many pairwise comparisons between subgroups were performed, which increases the risk of false-positive findings (i.e. type I errors). Even though we applied a strict *P* value correction for multiple testing, this naturally does not eliminate the risk of false-positive findings. The analyses on resistance gene and VG content should therefore be viewed as hypothesis generating.

In conclusion, associations between clinical characteristics of ECB episodes and molecular content of *E. coli* isolates were limited. However, we did identify important differences in clonality, serotypes, antimicrobial resistance genes and VG scores between non-ESBL-Ec and ESBL-Ec blood isolates that reached beyond their phenotypic ESBL-positivity. Future studies that aim to describe the molecular epidemiology of ECB should therefore preferably focus on *E. coli* without preselection on ESBL-positivity, to limit the risk of inferring characteristics of resistant *E. coli* to the *E. coli* population as a whole.

## Supporting information

**S1 Appendix. ST distribution in epidemiological subgroups.**
(PDF)

**S2 Appendix. Serotype distribution in epidemiological subgroups.**
(PDF)

**S3 Appendix. Supplementary data on acquired resistance gene content.**
(PDF)

**S4 Appendix. Supplementary data on virulence gene content.**
(PDF)

## Acknowledgments

We would sincerely like to thank Kim van der Zwaluw, Carlo Verhulst and Judith Vlooswijk for their contributions in the laboratory execution of the study. Preliminary results of this paper were presented at the 29th European Congress of Clinical Microbiology & Infectious Diseases, Amsterdam, the Netherlands, 13–16 April 2019 (P1450).

## Author Contributions

**Conceptualization:** Denise van Hout, Tess D. Verschuuren, Patricia C. J. Bruijning-Verhagen, Thijs Bosch, Marc J. M. Bonten, Jan A. J. W. Kluytmans.

**Data curation:** Denise van Hout, Tess D. Verschuuren, Thijs Bosch.

**Formal analysis:** Denise van Hout, Tess D. Verschuuren, Patricia C. J. Bruijning-Verhagen, Thijs Bosch, Marc J. M. Bonten, Jan A. J. W. Kluytmans.

**Funding acquisition:** Thijs Bosch, Marc J. M. Bonten.

**Investigation:** Denise van Hout, Tess D. Verschuuren, Thijs Bosch.

**Methodology:** Denise van Hout, Tess D. Verschuuren, Patricia C. J. Bruijning-Verhagen, Thijs Bosch, Anita C. Schürch, Rob J. L. Willems.

**Project administration:** Denise van Hout, Tess D. Verschuuren.

**Resources:** Thijs Bosch, Anita C. Schürch, Rob J. L. Willems, Marc J. M. Bonten, Jan A. J. W. Kluytmans.

**Software:** Denise van Hout, Tess D. Verschuuren, Thijs Bosch, Anita C. Schürch, Rob J. L. Willems.

**Supervision:** Patricia C. J. Bruijning-Verhagen, Thijs Bosch, Anita C. Schürch, Rob J. L. Willems, Marc J. M. Bonten, Jan A. J. W. Kluytmans.

**Validation:** Denise van Hout, Tess D. Verschuuren, Patricia C. J. Bruijning-Verhagen, Anita C. Schürch.

**Visualization:** Denise van Hout.

**Writing – original draft:** Denise van Hout.

**Writing – review & editing:** Denise van Hout, Tess D. Verschuuren, Patricia C. J. Bruijning-Verhagen, Thijs Bosch, Anita C. Schürch, Rob J. L. Willems, Marc J. M. Bonten, Jan A. J. W. Kluytmans.

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
