## [Decision Letter · Decision Letter 0]

13 Dec 2019

PONE-D-19-29528

Extended-spectrum beta-lactamase (ESBL)-producing and non-ESBL-producing Escherichia coli isolates causing bacteremia in the Netherlands (2014 – 2016) differ in clonal distribution, antimicrobial resistance gene and virulence gene content

PLOS ONE

Dear Mrs. van Hout,

Thank you for submitting your manuscript to PLOS ONE. After careful consideration, we feel that it has merit but does not fully meet PLOS ONE’s publication criteria as it currently stands. Therefore, we invite you to submit a revised version of the manuscript that addresses the points raised during the review process.

I am sorry for the delay with the revision process but it was not easy to find available expert reviewers. The manuscript is very good but please submit a revised version that addresses all the points raised by the two reviewers.  Furthermore, I agree that the manuscript is long. Please be more concise and provide clear explanations.

We would appreciate receiving your revised manuscript by Jan 27 2020 11:59PM. To enhance the reproducibility of your results, we recommend that if applicable you deposit your laboratory protocols in protocols.io, where a protocol can be assigned its own identifier (DOI) such that it can be cited independently in the future. For instructions see: http://journals.plos.org/plosone/s/submission-guidelines#loc-laboratory-protocols

We look forward to receiving your revised manuscript.

Kind regards,

Jose Melo-Cristino, M.D., Ph.D.

Academic Editor

PLOS ONE

Journal Requirements:

1.

Reviewers' comments:

Reviewer's Responses to Questions

**Comments to the Author**

1. Is the manuscript technically sound, and do the data support the conclusions?

Reviewer #1: Yes

Reviewer #2: Yes

2. Has the statistical analysis been performed appropriately and rigorously? 

Reviewer #1: Yes

Reviewer #2: I Don't Know

3. Have the authors made all data underlying the findings in their manuscript fully available?

Reviewer #1: Yes

Reviewer #2: Yes

4. Is the manuscript presented in an intelligible fashion and written in standard English?

Reviewer #1: Yes

Reviewer #2: Yes

5. Review Comments to the Author

Reviewer #1: This is a well-designed, and overall well performed study. Lots of data that have been obtained here, and Tables and Figures and well thought in order to make the whole set easy to understand and analyse. I would suggest to include S1, S2 and S3 in the main manuscript rather than as Supplements, but this is a detail somehow.

What is missing on my opinion to provide the reader all the tools to analyse those data in term of population structure distribution are some words about the links between ST. Indeed it is essential to figure out whether a so-called ST73 for instance is phylogenetically close or by contrast very different from an ST131. This would mean completely different things as you may guess.

The part dealing with virulence factors might be very much reduced on my opinion.

In fact, the entire manuscript would benefit from signifiant shortening, and to condense it would surely make it more attractive to the reader.

Reviewer #2: The manuscript describes a molecular and serologic epidemiological study of E. coli isolates from bacteremia ESBL and non- ESBL producers. The manuscript is well written, and despite not showing results completelly new, is interesting to a scientific reader and add more data on the epidemiology of E. coli recovered from blood samples.

The molecular study is very complete and WGS was used for analysing the genome of the isolates.

I just have a few remarks:

Line 276: some VG scores were similar, not completelly different (lines 250-59)

Line 309: It is well known that usually in E. coli there is an inversion in resistance genes versus virulence genes, mostly in UPEC (see Johnson et al papers). The exception is ST131, the most well studied ST right now. So, this is not a novelty.

Minor comment: Introduction, authors repeat consecutively references (1-4) from lines 62 until 69. Not necessary.

6. PLOS authors have the option to publish the peer review history of their article (what does this mean?). If published, this will include your full peer review and any attached files.

Reviewer #1: No

Reviewer #2: No

---

## [Author Response · Author response to Decision Letter 0]

16 Dec 2019

Dear Dr. José Melo-Cristino,

We sincerely thank the reviewers for their valuable time to evaluate this manuscript. The paper has been modified in response to the comments and suggestions. All changes are highlighted in the revised document in yellow. Furthermore, the manuscript was shortened by 651 words (20%). In the following we address all comments point by point. The response letter has also been submitted as a seperate PDF document. 

Yours sincerely, 

Denise van Hout

Response to Reviewer #1:

This is a well-designed, and overall well performed study. Lots of data that have been obtained here, and Tables and Figures and well thought in order to make the whole set easy to understand and analyse. 

Response: We would sincerely like to thank the reviewer for the time taken to critically appraise this manuscript and the input to further improve the manuscript. In the following we will address the comments point by point. 

I would suggest to include S1, S2 and S3 in the main manuscript rather than as Supplements, but this is a detail somehow.

Reply: Thank you for this suggestion. We have indeed considered to move some of the figures from the Supplement in to the main manuscript, given they provide additional illustrative information. However, given the manuscript is very data-dense, including 4 figures and 2 tables already, we are afraid addition of more figures might be a bit too much. We would therefore like to request the reviewer to keep the figures in the supplementary material. 

What is missing on my opinion to provide the reader all the tools to analyse those data in term of population structure distribution are some words about the links between ST. Indeed it is essential to figure out whether a so-called ST73 for instance is phylogenetically close or by contrast very different from an ST131. This would mean completely different things as you may guess.

Reply: Thank you for this comment. We indeed agree with the reviewer that the manuscript indeed lacked guidance on how to interpret the results of the different ST distributions. We have therefore added a comment to the Methods section that includes a short explanation of the MLST scheme (lines 101-102 of the revised manuscript) and have added a short comment to the Discussion section about the known phylogenetic relationship between ST73 and ST131, see lines 238-239 of the revised manuscript. 

The part dealing with virulence factors might be very much reduced on my opinion.

In fact, the entire manuscript would benefit from signifiant shortening, and to condense it would surely make it more attractive to the reader.

Reply: We agree with the reviewer that the part on virulence genes was too elaborate and have therefore shortened this paragraph by almost half (lines 209-216 of the revised manuscript). Furthermore, the rest of the manuscript was also shortened (by 20%), as suggested by the reviewer and the academic editor (mainly in the introduction, methods and in the description of results on resistance gene and VG content). 

Response to Reviewer #2:

The manuscript describes a molecular and serologic epidemiological study of E. coli isolates from bacteremia ESBL and non- ESBL producers. The manuscript is well written, and despite not showing results completelly new, is interesting to a scientific reader and add more data on the epidemiology of E. coli recovered from blood samples.

The molecular study is very complete and WGS was used for analysing the genome of the isolates.

Reply: We would like to thank the reviewer for the time and we sincerely appreciate the careful reading of our manuscript. In the following we respond to the comments point by point. 

I just have a few remarks:

Line 276: some VG scores were similar, not completelly different (lines 250-59)

Reply: Thank you for this comment. The reviewer is right to point out that when assessing epidemiological subgroups, most of the VG scores were not significantly different between groups. These analyses were stratified on ESBL-positivity, thus were performed for non-ESBL-Ec and ESBL-Ec separately. We would like to point out that the statement made in line 276 however (line 233 of the revised manuscript), does not refer to the epidemiological subgroups, but refers to the overall significant difference in VG score that was found between non-ESBL-Ec and ESBL-Ec isolates, as provided in lines 211-212 of the revised manuscript. The paragraph on VG scores has been shortened and hopefully is now more clear to the reader.

Line 309: It is well known that usually in E. coli there is an inversion in resistance genes versus virulence genes, mostly in UPEC (see Johnson et al papers). The exception is ST131, the most well studied ST right now. So, this is not a novelty.

Reply: We thank the reviewer for this comment; we agree that the paragraph did not appropriately reflect current literature and have revised it accordingly. This also included referencing of additional papers, among which important papers performed by Johnson et al., Vila et al. and Velasco et al. (lines 267-273 of the revised manuscript).

Minor comment: Introduction, authors repeat consecutively references (1-4) from lines 62 until 69. Not necessary.

Reply: Thank you for this suggestion, the changed were made as suggested by the reviewer.

---

## [Editor Report · Decision Letter 1]

20 Dec 2019

PONE-D-19-29528R1

Extended-spectrum beta-lactamase (ESBL)-producing and non-ESBL-producing Escherichia coli isolates causing bacteremia in the Netherlands (2014 – 2016) differ in clonal distribution, antimicrobial resistance gene and virulence gene content

PLOS ONE

Dear Mrs. van Hout,

Thank you for submitting your manuscript to PLOS ONE. After careful consideration, we feel that it has merit but does not fully meet PLOS ONE’s publication criteria as it currently stands. Therefore, we invite you to submit a revised version of the manuscript that addresses the points raised during the review process.

In Table 2, after some serotypes, you have indicated (%), as it is indicated after the value of N in the top of both vertical rows. Can you clarify the mention of (%) in some serotypes?

We would appreciate receiving your revised manuscript by Feb 03 2020 11:59PM. To enhance the reproducibility of your results, we recommend that if applicable you deposit your laboratory protocols in protocols.io, where a protocol can be assigned its own identifier (DOI) such that it can be cited independently in the future. For instructions see: http://journals.plos.org/plosone/s/submission-guidelines#loc-laboratory-protocols

We look forward to receiving your revised manuscript.

Kind regards,

Jose Melo-Cristino, M.D., Ph.D.

Academic Editor

PLOS ONE

---

## [Editor Report · Decision Letter 2]

26 Dec 2019

Extended-spectrum beta-lactamase (ESBL)-producing and non-ESBL-producing Escherichia coli isolates causing bacteremia in the Netherlands (2014 – 2016) differ in clonal distribution, antimicrobial resistance gene and virulence gene content

PONE-D-19-29528R2

Dear Dr. van Hout,

We are pleased to inform you that your manuscript has been judged scientifically suitable for publication and will be formally accepted for publication once it complies with all outstanding technical requirements.

With kind regards,

Jose Melo-Cristino, M.D., Ph.D.

Academic Editor

PLOS ONE
---

## [Editor Report · Acceptance letter]

31 Dec 2019

PONE-D-19-29528R2 

Extended-spectrum beta-lactamase (ESBL)-producing and non-ESBL-producing Escherichia coli isolates causing bacteremia in the Netherlands (2014 – 2016) differ in clonal distribution, antimicrobial resistance gene and virulence gene content 

Dear Dr. van Hout:

I am pleased to inform you that your manuscript has been deemed suitable for publication in PLOS ONE. Congratulations! Your manuscript is now with our production department. 

With kind regards,

on behalf of

Prof. Jose Melo-Cristino 

Academic Editor

PLOS ONE